# Loss of presenilin function is associated with a selective gain of APP function

Carole Deyts, Mary Clutter, Stacy Herrera, Natalia Jovanovic, Anna Goddi, Angèle T Parent*

Departments of Neurobiology, The University of Chicago, Chicago, United States

**Abstract** Presenilin 1 (PS1) is an essential γ-secretase component, the enzyme responsible for amyloid precursor protein (APP) intramembraneous cleavage. Mutations in PS1 lead to dominant-inheritance of early-onset familial Alzheimer's disease (FAD). Although expression of FAD-linked PS1 mutations enhances toxic Aβ production, the importance of other APP metabolites and γ-secretase substrates in the etiology of the disease has not been confirmed. We report that neurons expressing FAD-linked PS1 variants or functionally deficient PS1 exhibit enhanced axodendritic outgrowth due to increased levels of APP intracellular C-terminal fragment (APP-CTF). APP expression is required for exuberant neurite outgrowth and hippocampal axonal sprouting observed in knock-in mice expressing FAD-linked PS1 mutation. APP-CTF accumulation initiates CREB signaling cascade through an association of APP-CTF with Gαs protein. We demonstrate that pathological PS1 loss-of-function impinges on neurite formation through a selective APP gain-of-function that could impact on axodendritic connectivity and contribute to aberrant axonal sprouting observed in AD patients.

*For correspondence: aparent@ uchicago.edu

Competing interests: The authors declare that no competing interests exist.

## Introduction

Alzheimer's disease (AD) is a progressive neurodegenerative disease pathologically characterized by a cerebral deposition of β-amyloid peptide (Aβ) in senile plaques and neuronal loss. Inheritance of dominant forms of mutations in genes encoding amyloid precursor protein (APP) and presenilins (*PSEN1* and *PSEN2* genes) cause aggressive forms of early onset familial AD (FAD). A mutation in presenilin genes causes autosomal dominant early-onset familial Alzheimer's disease (FAD) (*Tanzi and Bertram, 2001*). Gene knockout studies in mice reveal that PS1 acts as the catalytic core of the multisubunit γ-secretase complex that is responsible for regulated intramembranous proteolysis of several type-I transmembrane protein substrates. Over 90 substrates have been identified so far including amyloid precursor protein (APP), Notch and Eph receptors and ligands, cadherins, and deleted in colorectal cancer (DCC) (*Haapasalo and Kovacs, 2011*; *Kopan and Ilagan, 2004*; *McCarthy et al., 2009*; *Parks and Curtis, 2007*). Several of these substrates are known for their diverse functions during neuronal development including axon guidance, neurite outgrowth, and synaptogenesis. In the case of APP, the most studied γ-secretase substrate, sequential ectodomain shedding by α-secretase, which occurs mainly at the cell surface, or β-secretase is required before subsequent cleavage by γ-secretase (*Deyts et al., 2016*; *Haass et al., 2012*; *Thinakaran and Koo, 2008*). Therefore, cleavage of full-length APP (APP-FL) by α- or β-secretases releases the entire ecto-domain (soluble APPα or soluble APPβ, respectively), leaving behind membrane-bound C-terminal fragments (CTF), comprising the transmembrane and cytoplasmic domain (APP-C83 and APP-C99, respectively). Subsequent cleavage of APP-CTF by γ-secretase releases the cytosolic domain from the membrane (APP intracellular domain, AICD) and either p3 fragment or the neurotoxic Aβ peptide. Consequently, inhibiting γ-secretase activity would prevent accumulation of Aβ and AICD in

**eLife digest** One of the hallmarks of Alzheimer's disease is the accumulation within the brain of sticky deposits called plaques. These plaques form from clumps of molecules called amyloid-beta peptide. An enzyme called gamma-secretase generates the amyloid-beta peptide, by cutting it from a membrane-associated protein called APP. This enzyme consists of multiple subunits, and a mutation in one of these – presenilin-1 – causes a particularly severe form of Alzheimer's disease.

For decades, research into Alzheimer's disease has focused on the harmful effects of amyloid-beta peptides and plaques. However, Deyts et al. now argue that the protein that gives rise to amyloid-beta peptides has a more direct role in Alzheimer's disease than previously thought. Specifically, APP may contribute to the harmful effects of the presenilin-1 mutations.

By studying genetically modified mice carrying a human presenilin-1 mutation, Deyts et al. show that some of these animals' nerve cells grow abnormally. Their cell bodies sprout too many branches, while their nerve fibers – which carry electrical signals away from the cell body – become too long. These abnormalities resemble changes seen in the brain in Alzheimer's disease. Unexpectedly, however, deleting the gene for APP in the presenilin-1 mutant mice prevents the changes from occurring. This suggests that APP must be present for the presenilin-1 mutation to exert this unwanted effect.

An increase in APP-driven signaling within cells seems to trigger the observed abnormalities in nerve cells. The presenilin-1 mutation modifies how gamma-secretase cuts APP at the cell membrane to produce amyloid-beta peptides. This frees up the APP to instead interact with signaling cascades inside the cell. Given that gamma-secretase is a key therapeutic target in Alzheimer's disease, further work is needed to explore the implications of these protein interactions for potential treatments.

favor of accumulation of APP-C83 or APP-C99 depending on their prior non-amylogenic α-CTF or amylogenic β-CTF cleavage products, respectively.

It has been proposed that γ-secretase complex activity may serve as the membrane proteasome that removes C-terminal stubs generated after ectodomain shedding and prevents further cell-surface signaling by a variety of substrates (*Kopan and Ilagan, 2004*). No evidence has been described that such a role may be an important contributor of disease states associated with γ-secretase components. The contribution of APP holoprotein and other PS1-dependent substrates in the AD etiology has not been fully examined. The present work focuses on determining the importance of PS1-dependent modulation of APP-CTF accumulation and the subsequent effects on associated signaling that promote neurite outgrowth. We report here that alteration of γ-secretase activity, through pharmacologic PS1 inhibition, genetic *Psen1* ablation, or expression of FAD-linked PS1 variants, enhances neurite outgrowth. Our findings indicate that APP is required for PS1-dependent changes in neurite outgrowth. Ablation of APP expression prevented aberrant axonal sprouting observed in the hippocampal dentate gyrus of *PSEN1* knock-in mouse model harboring FAD-linked PS1 variant. APP-CTF accumulation contributed to γ-secretase-dependent increases of CREB signaling cascade seen in neurons that exhibit PS1 loss-of-function, an effect that was prohibited through adenylate cyclase inhibition. Our results provide the first demonstration that a pathological loss of PS1 function leads to a selective gain of APP function that may impact axodendritic connectivity.

## Results

### Exuberant axodendritic outgrowth associated with γ-secretase inhibition requires APP expression

We previously demonstrated that accumulation of APP-CTF in the raft produced a marked increase of neurite extension in a variety of neuronal cells including cortical neurons (*Deyts et al., 2012*). We also observed that overexpression of APP and concurrent γ-secretase inhibition, that produce an accumulation of APP-CTF and other γ-secretase-dependent CTF substrates, leads to an increase in neurite outgrowth (*Deyts et al., 2012*). Given the fact that APP plays important roles in neurite

outgrowth through the function of its extra- and intracellular fragments (review by (*Chasseigneaux and Allinquant, 2011*; *Muller and Zheng, 2012*; *Nicolas and Hassan, 2014*)), we wanted to establish the physiological significance of endogenous APP and its metabolites that are generated through modulation of γ-secretase activity. First, we have examined neurite outgrowth in cortical neurons lacking APP using APP knockout (APP KO) mice (*Figure 1*). Lack of APP expression modestly (but significantly) and selectively altered axonal outgrowth (*Figure 1b1*; WT, 1.00 ± 0.03; APP KO, 0.77 ± 0.05, *P*<0.001), whereas the dendritic outgrowth was not affected (*Figure 1b2*, see also *Figure 1—figure supplement 1*). Second, we confirmed the physiological consequence of γ-secretase inhibition on neurite outgrowth, using γ-secretase inhibitor Compound E (*Seiffert et al., 2000*). We observed that inhibition of γ-secretase in WT neurons enhances axonal and dendritic outgrowth (*Figure 1b1 and b2*), with a more predominant effect on the axon as reflected by an increase of the axon/dendrite area ratio (see *Figure 1b3*; WT, 1.78 ± 0.07; WT+CompE, 2.52 ± 0.18, *P*< 0.05). Next, we examined whether γ-secretase activity influenced neurite outgrowth in neurons lacking APP. Interestingly, we observed that neurons treated with γ-secretase inhibitor did not exhibit axonal or dendritic outgrowth in the absence of APP expression (*Figure 1b1 and b2*), suggesting that APP is the main contributing factor in γ-secretase-mediated neurite outgrowth.

We subsequently decided to overexpress APP in an attempt to reinstate the effect seen in APP KO neurons to the WT level. We observed that concomitant expression of APP-FL and inhibition of γ-secretase are sufficient to reestablish the enhancement of axodendritic outgrowth at the level seen in WT neurons treated with a γ-secretase inhibitor (*Figure 1a and b*). In our neuronal cultures, fractionation of lysates on Tris-Tricine gels showed that the inhibition of γ-secretase activity produced an overall increase of APP-CTF whereas APP-FL was not changed (*Figure 1c2* and *Figure 1—figure supplement 2*). As expected, we confirmed that γ-secretase dependent processing of APP affects accumulation of both APP-CTFα and APP-CTFβ, at least in HEK293 and COS-7 cells (*Figure 1—figure supplement 2*). Overexpression of APP-FL was not sufficient to induce axodendritic outgrowth, although accumulation of APP-CTF through γ-secretase inhibition did promote axodendritic outgrowth. This set of data confirmed that inhibition of γ-secretase activity favors a preferential augmentation of axonal outgrowth in comparison to the dendritic compartment, as shown by the increase of the axon/dendrite ratio (*Figure 1b3*). This effect was abolished in the absence of endogenous APP expression and restored by APP overexpression. Our findings emphasize that both γ-secretase inhibition and APP expression are required to induce axodendritic outgrowth.

## Expression of FAD-linked PS1 variants promotes neurite outgrowth

It is well established that inhibition of γ-secretase activity, or lack of presenilin 1 (PS1) expression, produces an endogenous accumulation of APP-CTF in the brain, neuronal cultures, or cell lines (*Bentahir et al., 2006*; *De Strooper et al., 1998*; *Parent et al., 2005*; *Walker et al., 2005*; *Woodruff et al., 2013*). In earlier studies, it has been proposed that expression of FAD-linked PS1 mutant proteins may attenuate γ-secretase activity, consequently limiting proteolysis of full-length protein substrates and, therefore, accumulating their CTF (*Bentahir et al., 2006*; *De Strooper, 2007*; *Shen and Kelleher, 2007*; *Walker et al., 2005*; *Wolfe, 2007*; *Woodruff et al., 2013*). Based on our results reported in *Figure 1*, we wondered whether pathogenic PS1 mutations, that have been described to affect γ-secretase activity, might influence neurite outgrowth. We focused on three PS1 variants that carry a PS1-M146L substitution and PS1-ΔE9 deletion of exon 9, which is known to cause earlier age-of-onset FAD, and a known dominant negative PS1-D385A substitution. We stably transfected HEK293 cells with wild-type human PS1 (PS1-WT) or with PS1 mutations. As shown in *Figure 2a1*, pools of established stably transfected HEK293 cells expressed comparable levels of PS1 protein, as shown by cell lysates immunoblotted with a PS1 polyclonal antibody that detects PS1-FL and PS1-NTF. Next, we analyzed the ability of PS1 mutation to modulate the accumulation of APP-CTF in PS1 stable HEK293 cells that overexpressed APP-FL (*Figure 2a2*). We observed that APP-CTF accumulates in PS1 mutants, as compared to PS1-WT (*Figure 2a2*), confirming previous findings that pathogenic PS1 mutations might reduce γ-secretase activity (*Bentahir et al., 2006*; *De Strooper, 2007*; *Heilig et al., 2013*; *Shen and Kelleher, 2007*; *Walker et al., 2005*; *Woodruff et al., 2013*; *Xia et al., 2015*). Interestingly, we observed that neurons expressing PS1-ΔE9 mutation exhibit a significant increase in axodendritic outgrowth (*Figure 2b and c*), an effect that is exacerbated by the presence of APP-FL in neurons overexpressing PS1-M146L and PS1-D385A mutations. Taken together, our results demonstrate that pathogenic

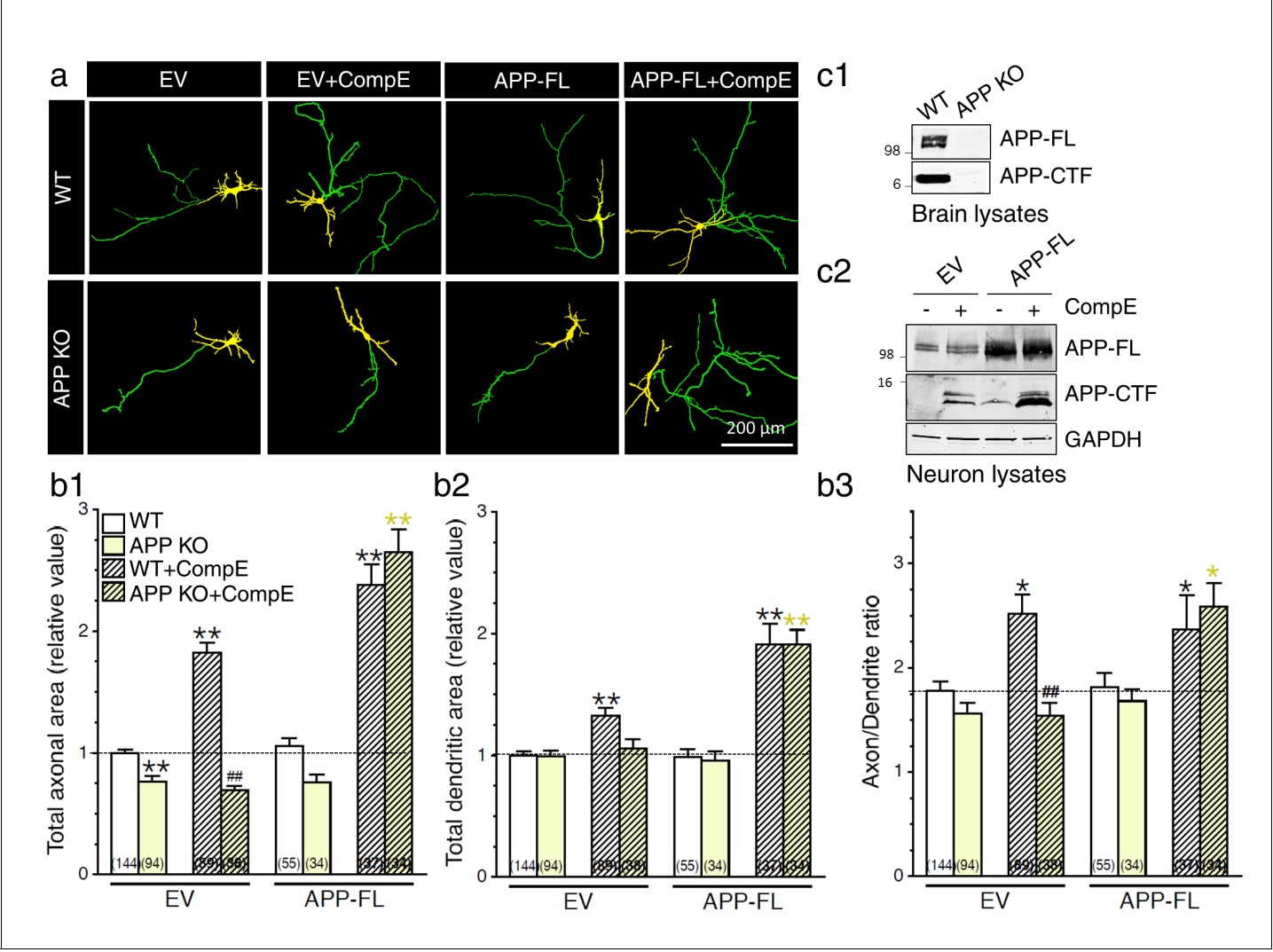

**Figure 1.** Exuberant axodendritic outgrowth associated with γ-secretase inhibition requires APP expression. (a) WT or APP KO primary cortical neurons (8 DIV) coexpressing YFP and EV or APP-FL were treated with Compound E (CompE 10 nM, 24 hr) and immunostained with MAP2 antibody. (a) Representative overlay images of YFP and MAP2 staining reveal axon (green) and dendrites (yellow). (b) Quantitative analysis of neurite outgrowth is represented as relative changes in total axonal area (b1), dendritic area (b2), and axon/dendrite ratio (b3) in treated or untreated neurons with Compound E, expressing either EV or APP-FL as compared to WT-EV control group. (c1) Endogenous APP full-length (APP-FL) and accumulation of APP-CTF were detected by immunoblotting brain lysates from WT and APP KO mice with CTM1 antibody, which recognizes the C-terminus of APP (*von Koch et al., 1997*). (c2) Accumulation of APP-CTF in WT neurons overexpressing empty vector (EV) or APP-FL is detected by Western blot using a Tris-Tricine gel, before or after treatment with Compound E (CompE: 10 nM, 24 hr). GAPDH was used as loading control. Statistical analysis was performed using ANOVA Kruskal-Wallis test followed by Dunn's post hoc multiple comparison analysis. *p<0.05, **p<0.001 compared to WT-EV; yellow *p<0.05, **p<0.001 compared to APP KO, and ##p<0.001 compared to WT treated with CompE. The total number of quantified cells is shown in parentheses (WT, n = 13 embryos; APP KO, n = 6 embryos; Compound E, n = 5 embryos). Error bars indicate SEM.

The following figure supplements are available for figure 1:

**Figure supplement 1.** Changes in axodendritic outgrowth associated with γ-secretase inhibition requires APP expression.

**Figure supplement 2.** Identification of APP-CTF through differential processing of APP by secretases.

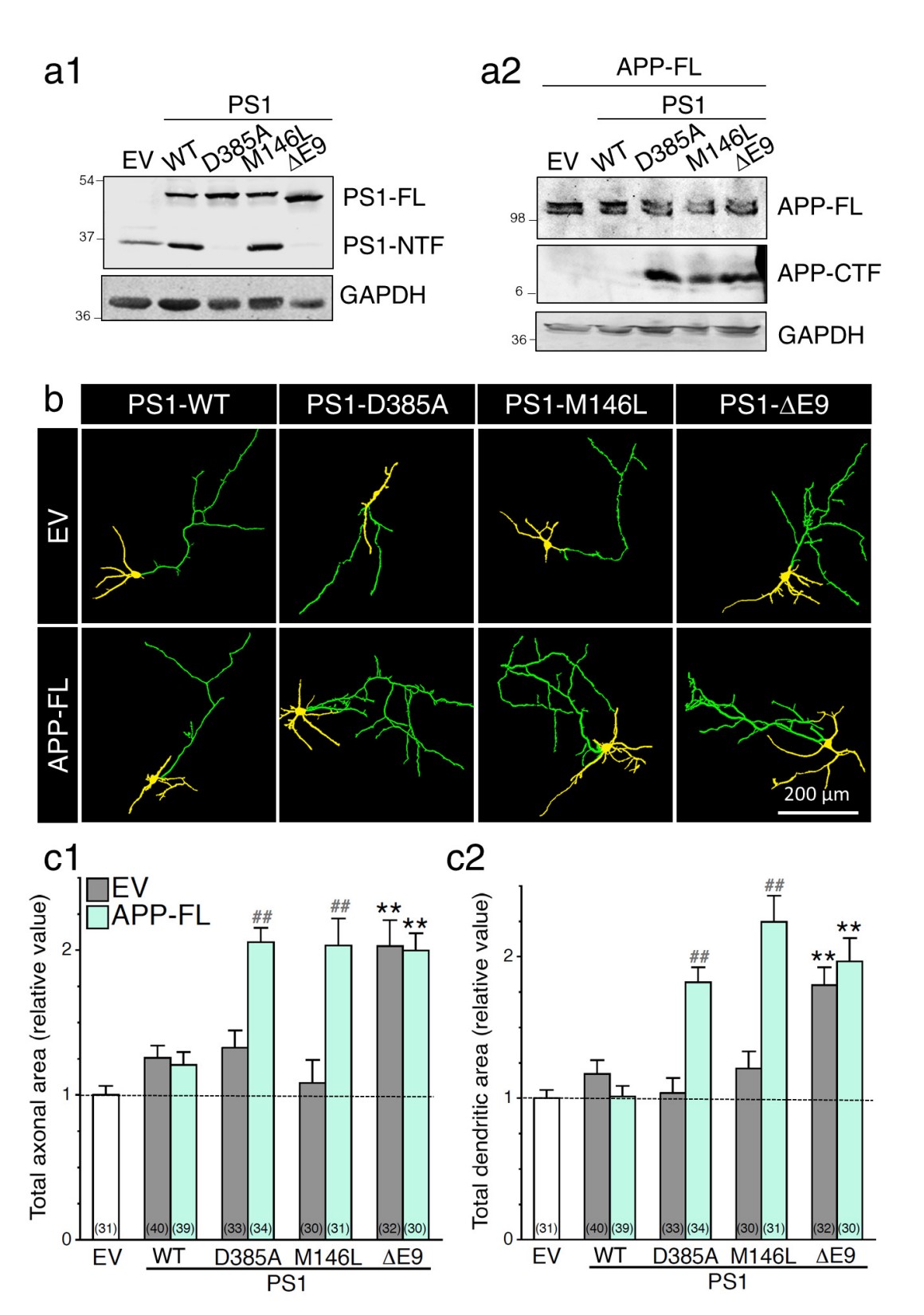

**Figure 2.** Expression of FAD-linked PS1 variants promotes neurite outgrowth. FAD-linked PS1 mutations enhance axonal and dendritic arborization that correlates with APP C-terminal fragment (APP-CTF) accumulation. FAD-linked PS1 mutations affect APP processing in cells expressing APP full-length

*Figure 2 continued on next page*

*Figure 2 continued*

(APP-FL). (a) Stable HEK293 cells overexpressing FAD-linked PS1 variants (a1) were transiently transfected with APP-FL (a2). (a1) The PS1$_{NT}$ polyclonal antibody was used to detect PS1 full-length (PS1-FL) and PS1 N-terminal fragment (PS1-NTF). (a2) The CTM1 polyclonal antibody was used to detect APP-FL and APP-CTF accumulation. (b) GAPDH was used as loading control. (b) WT primary cortical neurons (8 DIV) coexpressing YFP, APP-FL and PS1 mutants were immunostained with MAP2 antibody. Representative overlay images of YFP and MAP2 staining reveal axonal (green) and dendritic (yellow) arbors. (c) Quantification of the total axonal (b1) and dendritic (b2) areas is shown in cortical neurons (7–8 DIV) 24 hr following transfection of PS1 variants (in gray) with APP-FL (in light green). The total number of quantified cells is shown in parentheses (n = 5 embryos for each transfected condition). Error bars indicate SEM.

PS1 mutations favor axodendritic outgrowth as seen under a condition that reduces γ-secretase activity.

## Neurons expressing FAD-linked PS1 mutations exhibit partial loss of γ-secretase activity associated with APP-dependent increases of neurite outgrowth

To substantiate the impact of PS1 function on γ-secretase activity, we took advantage of knockout and knock-in mouse models with targeted deletion of *Psen1* (PS1 KO) and replaced *Psen1* gene with human FAD-linked *PSEN1-M146V* variant (PS1 KI), respectively. In this later strain, PS1-M146V protein is expressed at a level equivalent to WT protein level, therefore mimicking human FAD patients carrying this mutation. These PS1 KI mice do not exhibit overt Aβ accumulation or deposition; thus, the main interest of using this mouse model is to understand the contribution of early development of AD process. As shown in *Figure 3a*, Western blot analysis of steady-state APP expression levels in brains of E15 mouse embryos reveals a significant increase of APP-CTFα in PS1 KI, and larger increase of both APP-CTFα and APP-CTFβ in PS1 KO as compared to WT. Analysis of brain lysates revealed similar increases of APP-CTF in total membrane fractions of heterozygous (PS1 KI/+) and homozygous (PS1 KI/KI) brains harvested during synaptogenesis (*Figure 3b* and *Figure 3—figure supplement 1a1*, postnatal day 10 – P10) or in adulthood (*Figure 3—figure supplement 1*; 6 months-old – 6M), as compared to their WT littermates. Levels of APP-FL were not significantly affected among these groups (*Figure 3b* and *Figure 3—figure supplement 1a*). Consequently, ratios of APP-CTF/APP-FL were increased in PS1 KI brains (*Figure 3b2* and *Figure 3—figure supplement 1b2*), suggesting that the FAD-linked PS1 mutation may result in loss of activity toward APP proteolysis, leading to accumulation of membrane-tethered APP-CTF. However, APP-CTF accumulated to a lesser extent in PS1 KI as compared to PS1 KO (*Figure 3a*), therefore supporting the notion of partial loss-of-function mutant.

Next, we investigated the changes of γ-secretase activity on axodendritic development using primary neuronal cultures generated from PS1 KO and PS1 KI mice and their WT littermates. We observed a significant increase in both axonal and dendritic outgrowth in PS1 KI cultured neurons as compared to WT, an effect as substantial as the effect seen in neurons lacking PS1 expression (*Figure 3c and d*). Axonal and dendritic outgrowth was increased without apparent bias toward either neuronal compartment. Strikingly, we noticed that PS1 mutation-induced neurite outgrowth was comparable in neurons expressing one or two copies of the mutated gene, which is consistent with the idea of autosomal dominant inheritance of *PSEN1* mutations. Our findings also suggest that modest reduction in γ-secretase activity through an endogenous level of PS1 mutation expression is sufficient to influence drastically axodendritic outgrowth, an effect that emerges in conjunction with modest APP-CTF accumulation. To follow-up, we investigated the influence of APP expression on neurite extension. We examined whether neurons lacking APP expression would exhibit a similar level of exuberant neurite outgrowth in FAD-linked PS1-M146V variant using double crossed APP KO x PS1 KI mice. Remarkably, lack of APP expression abolished excess axonal and dendritic outgrowth found in neurons carrying the PS1-M146V mutation (*Figure 3d1*; 97% and 62% reduction, respectively). We also subjected neuronal culture generated from PS1 KI and APP KO x PS1 KI mice to γ-secretase inhibitor Compound E. We observed that presenilin-dependent enhancement of axodendritic outgrowth was not furthermore increased in APP KO x PS1 KI cultures (*Figure 3—figure supplement 2*), supporting the idea that APP is the major contributing substrate in FAD-linked PS1 mutation induced neurite outgrowth.

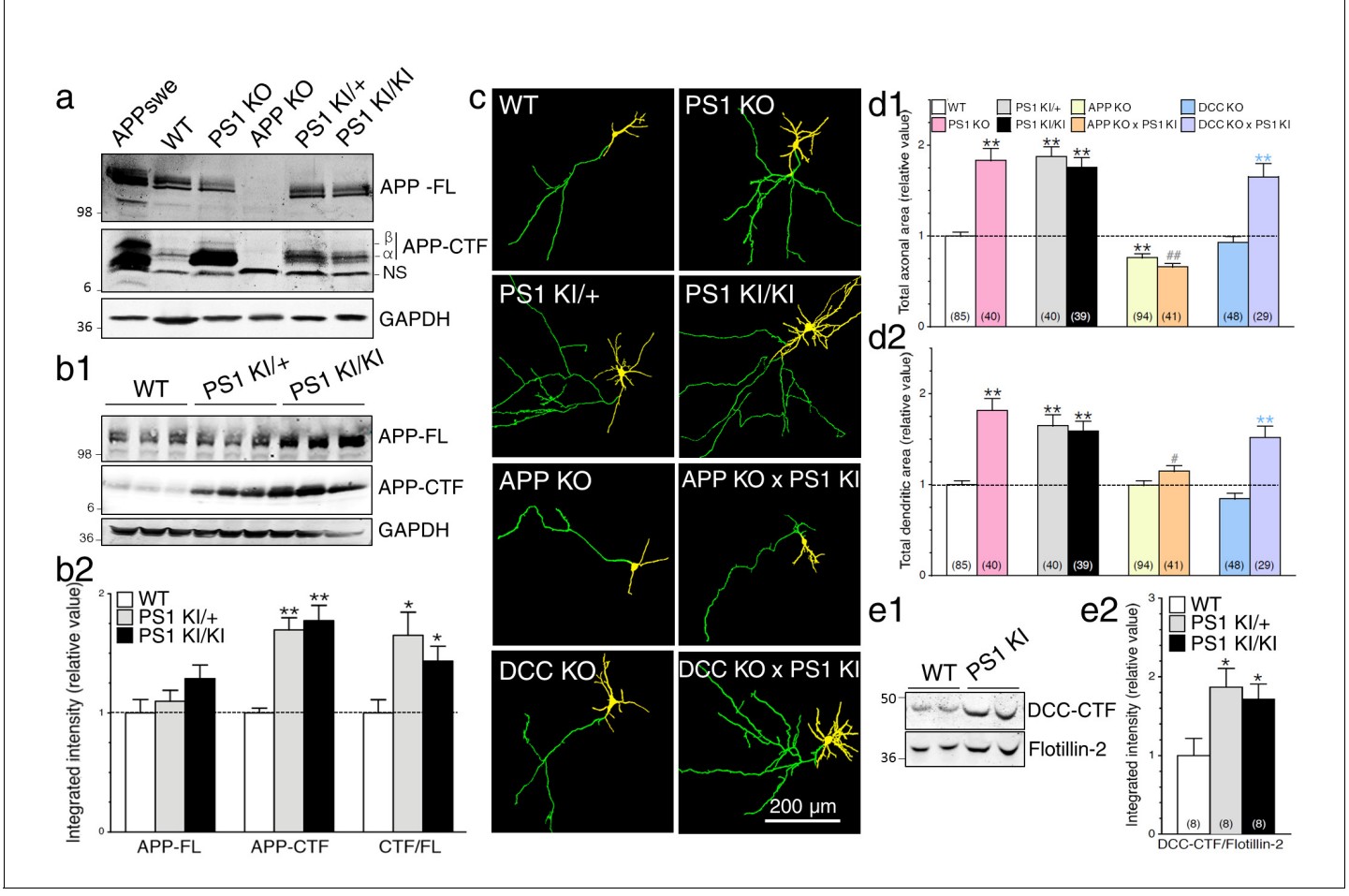

**Figure 3.** Neurons expressing FAD-linked PS1 mutations exhibit partial loss of γ-secretase activity associated with APP-dependent increases of neurite outgrowth. (a) Levels of APP full-length (APP-FL) and proteolytic C-terminal fragments APP-CTF (APP-CTFα and APP-CTFβ) accumulation were detected by high resolution Tris-Tricine Western blot analysis of lysates generated from WT, PS1 KO, APP KO, PS1 KI/+, and PS1 KI/KI mouse brains using CTM1 antibody. Lysate from HEK cells overexpressing APPswe and GAPDH antibody detection were used as control. Half amount of proteins was loaded for PS1 KO brain lysate. NS indicates a non-specific cross-reacting band. (b1) Endogenous APP-FL and accumulation of APP-CTF were detected with CTM1 antibody by immunoblotting brain lysates from postnatal day 10 (P10) transgenic *PSEN1-M146V* knock-in mice (PS1 KI). (b2) Quantitative analysis of APP-FL, APP-CTF accumulation and the ratio APP-CTF/APP-FL in PS1 KI is shown. Values are reported as a relative change in the intensity of the protein as compared with the WT littermates. (c) Primary cortical neurons (7–8 DIV) from WT, PS1 KO, PS1 KI (heterozygous PS1 KI/+ and homozygous PS1 KI/KI), APP KO, APP KO x PS1 KI, DCC KO and DCC KO x PS1 KI mice were transfected with YFP and immunostained with MAP2 antibody. Representative resulting overlay images reveal differences in axons (green) and dendrites (yellow). (d) Quantitative analysis of total axonal area (d1) and dendritic area (d2) are shown. Results are reported as relative values as compared to WT. (e) Western blot analysis of P10 brain lysates was performed to detect endogenous DCC-CTF fragments from WT and PS1 KI. (e1) Representative immunoblot lysate samples are shown. (e2) Quantitative analysis is shown as a relative change in the intensity of DCC-CTF expression as compared to WT littermates using Flotillin-2 as a loading control. Statistical analysis was performed using ANOVA Kruskal-Wallis test followed by Dunn's post hoc multiple comparison analysis. *p<0.05, **p<0.001 compared to WT, blue **p<0.001 compared to DCC KO, and ##p<0.001 compared to PS1 KI. The total number of neurons (from at least 3 independent sets of cultures) used for quantification is shown in parentheses (WT, n = 6 embryos; PS1 KO, n = 5 embryos; PS1 KI/+, n = 6 embryos; PS1 KI/KI, n = 6 embryos; APP KO, n = 6 embryos; APP KO x PS1 KI, n = 6 embryos; DCC KO, n = 7 embryos; DCC KO x PS1 KI, n = 5 embryos). Error bars indicate SEM.

The following figure supplements are available for figure 3:

**Figure supplement 1.** Intramembraneous proteolysis of APP in knock-in mice expressing FAD-linked PS1 variant.

**Figure supplement 2.** No additive effect of γ-secretase inhibition on neurite outgrowth in APP-deficient neurons expressing PS1 mutation.

**Figure supplement 3.** DCC is not an essential substrate in PS1-induced neurite outgrowth.

*Figure 3 continued on next page*

*Figure 3 continued*

**Figure supplement 4.** Intramembraneous proteolysis of DCC in brains of FAD-linked PS1 mutant knock-in mice.

To corroborate our finding on the predominant role of APP in neurite outgrowth, we investigated whether other γ-secretase substrates may contribute to or exacerbate the aberrant neuritic processes seen in neurons expressing PS1 mutation. We and others have previously reported that γ-secretase-dependent accumulation of Deleted Colorectal Cancer C-terminal fragment (DCC-CTF) promotes neurite outgrowth in neuroblastoma cells (*Parent et al., 2005*) and motor neuron explants (*Bai et al., 2011*). We used primary neurons generated from DCC KO mice, which lack DCC expression, to investigate the contribution of DCC to γ-secretase mediated axodendritic outgrowth. We observed that lack of DCC expression does not affect axonal or dendritic outgrowth in our cell culture system (*Figure 3c and d*). Within the same experimental conditions, sets of neurons were treated with γ-secretase inhibitor Compound E to abrogate substrate proteolysis. In these groups, enhancement of axodendritic outgrowth was similar in naïve WT as compared to DCC KO neurons treated with a γ-secretase inhibitor (*Figure 3—figure supplement 3*). We also confirmed that lack of DCC expression did not affect axodendritic outgrowth in neurons expressing FAD-linked PS1-M146V mutation (*Figure 3c and d*), even though DCC-CTF was increased in cultured neuron lysates of homozygous and heterozygote PS1 KI relative to WT (*Figure 3e*). An increase of DCC-CTF detection was also significant in vivo during early brain development and the period of synaptogenesis (E15-P14) in PS1 KI mice as compared to WT (*Figure 3—figure supplement 4*). Therefore, our results indicate that DCC is not an essential substrate for γ-secretase induced neurite outgrowth.

## APP-dependent axonal sprouting in the hippocampus of knock-in mice expressing FAD-linked PS1 variant

To examine the in vivo consequence of neurite outgrowth that we see in our neuronal culture system, we measured axonal sprouting in brain slices from 6 month-old animals using GAP-43 as a marker of axonal plasticity (reviewed by (*Benowitz and Routtenberg, 1997*; *Denny, 2006*)). As observed by several groups (*Benowitz et al., 1988*; *Goslin et al., 1988*; *Oestreicher and Gispen, 1986*), GAP-43 staining correlates selectively with axonal development and is prominent in brain areas where axon fibers project. Consistent with our neurite outgrowth analysis, GAP-43 expression is significantly increased in hippocampal areas of PS1 KI mice, an effect that was abrogated if APP expression was absent (see *Figure 4a*). Quantitative analysis reveals that intensity of GAP-43 immunostaining was selectively increased in the stratum lacunosum moleculare (LMol) and the outer and inner molecular layers of the dentate gyrus (OMoDG and IMoDG, respectively) in PS1 KI mice as compared to WT (*Figure 4b*). An increase of GAP-43 immunostaining intensity levels was seen in these selective areas as shown by representative line scan projection of GAP-43 staining in 6 month-old PS1 KI mice as compared to their WT littermates, an effect that was not observed in mice lacking APP expression (*Figure 4a3*). We confirmed by Western blot analysis that GAP-43 expression was increased as well in the hippocampus as early as at postnatal day 10 (P10) in PS1 KI as compared to APP KO x PS1 KI mice, an effect that was not observed in the cortex of these mice (*Figures 4b and c*). Confocal microscopy corroborated the existence of stronger immunofluorescence intensity in the dentate gyrus, especially in OMoDG layer of mice expressing FAD-linked PS1 variant as compared to their counterpart APP KO x PS1 KI littermates (*Figure 4d*). The surface render projection of zoomed areas revealed that GAP-43 immunostaining is more abundant in PS1 KI surrounding neurite-like structures, which was not seen in APP KO x PS1 KI mouse brain sections. Altogether, our results indicate that mice expressing an endogenous level of FAD-linked PS1 variant exhibit exuberant neurite outgrowth and axonal sprouting at close proximity to the perforant path, which requires APP expression.

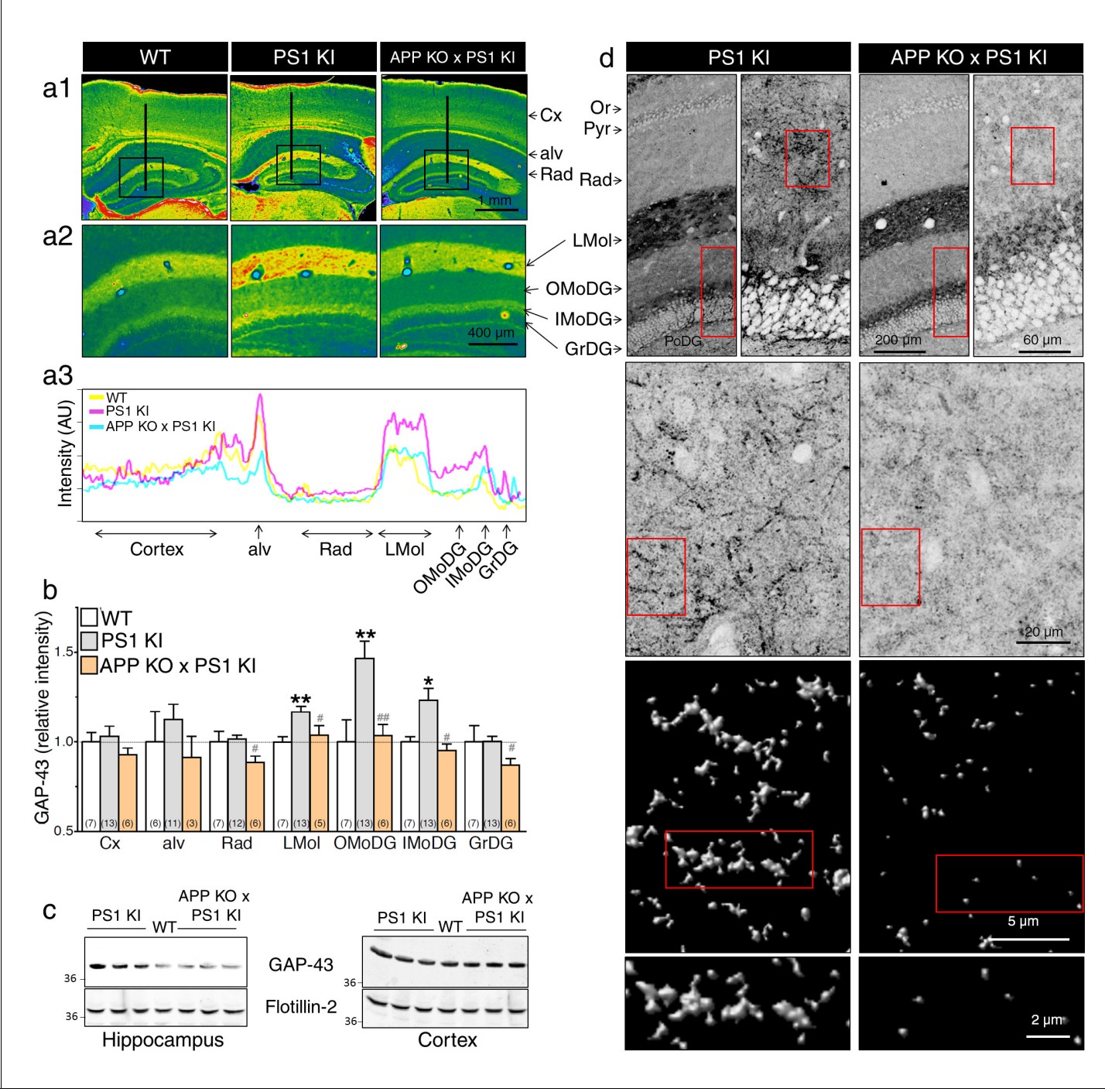

**Figure 4.** APP-dependent axonal sprouting in the hippocampus of knock-in mice expressing FAD-linked PS1 variant. (a) Immunohistochemistry of GAP-43 on coronal brain sections of 6 months-old WT, PS1 KI, and APP KO x PS1 KI mice is shown. (a1) Representative pseudocolor images of brain sections present GAP-43 staining intensity in several brain areas (Cx: cortex, alv: alveus, Or: oriens, Pyr: pyramidal layer, Rad: radiatum, LMol: lacunosum moleculare, OMoDG: outer molecular layer of the dentate gyrus, IMoDG: inner molecular layer of the dentate gyrus, GrDG: granular layer of the dentate gyrus, PoDG: polymorph layer of the dentate gyrus). (a2) Enlarged views of the dentate gyrus areas are shown. (a3) Representative line-scans (as indicated in a1) show levels of overlapping-peak intensity of GAP-43 immunostaining across cortical and hippocampal areas. More intense staining is noticeable especially in hippocampal areas of PS1 KI as compared to WT and APP KO x PS1 KI. (b) Quantitative analysis of GAP-43 staining intensity in several brain areas is represented as relative changes compared to WT. (c) Western blot analysis of steady-state levels of GAP-43 is examined using cortex and hippocampus lysates from WT, PS1 KI and APP KO x PS1 KI mouse brains taken at postnatal day 10 (P10). Flotillin-2 was used as loading control. (d) Confocal images of GAP-43 staining in 6 months-old mice showing the axonal sprouting in OMoDG layer of PS1 KI as compared to APP KO

*Figure 4 continued on next page*

*Figure 4 continued*

x PS1 KI mice. The boxed regions are shown as enlarged insets. Inverted images at the bottom show the surface rendered zoomed areas generated with Huygens software. Statistical analysis was performed using ANOVA Kruskal-Wallis test followed by Dunn's post hoc multiple comparison analysis. *p<0.05, **p<0.001 compared to WT littermates, and #p<0.05, ##p<0.001 compared to PS1 KI littermates. The total number of animals used for quantification is shown in parentheses. Error bars indicate SEM.

## Loss of γ-secretase activity is associated with a gain of APP-CTF signaling

We previously reported that accumulation of APP-CTF caused by overexpression of membrane-tethered APP intracellular domain (mAICD) favors axodendritic arborization as a result of direct coupling with Gα$_S$ and subsequent activation of adenylate cyclase (*Deyts et al., 2012*). Consequentially, cAMP/PKA cascade is initiated and leads to phosphorylation of CREB. As described in our previous studies (*Deyts et al., 2012*) and shown in *Figure 5*, we observed that concomitant expression of APP-FL and inhibition of γ-secretase activity favors neurite outgrowth (*Figure 5a*) and associated increase of phosphorylated CREB (pCREB, *Figure 5b*) in cortical neuronal cultures. In support of this finding, we also observed that neurons overexpressing APP-C99 (the APP β-secretase cleaved CTF by-product) exhibit increased axodendritic outgrowth that was prevented by the expression of a C99 construct lacking the Gα$_S$-protein interacting site where three alanine substitutions were introduced to replace the basic residues of the BBXXB motif in APP-CTF (named C99mutAAA; see *Figure 5a*). As shown in *Figure 5b*, we observed that neurons expressing APP-C99 or mAICD constructs exhibit a strong increase of CREB signaling illustrated by the enhancement of pCREB immunostaining. These outcomes were abolished in neurons overexpressing mutant lacking Gα$_S$-protein interacting site C99mutAAA and mAICDmutAAA, respectively (*Figure 5b1 and b2*). As expected for CREB signaling, these effects are more noticeable in the somatic/nuclear area as compared to dendritic projections (*Figure 5—figure supplement 1*). To further establish the importance of γ-secretase dependent APP-CTF accumulation in this process, we overexpressed membrane-targeted mAICD and APP-NTF by-products in primary mouse embryonic fibroblasts (MEF) or neurons generated from APP KO (*Figure 5c1 and c2*, respectively). We observed that neither sAPPα nor sAPPβ overexpression influenced CREB signaling in HEK293 cells or in neurons lacking APP expression (*Figure 5—figure supplement 2a* and *Figure 5c1*, respectively). We confirmed this observation by Western blots of MEF lysates generated from APP KO mice (*Figure 5c2*) and lysates of HEK293 cells (*Figure 5—figure supplement 2b*). Altogether, our results point out that increase in APP-CTF accumulation is causing changes in neurite outgrowth and CREB signaling in PS1 loss-of-function neurons, therefore supporting the idea that APP-CTF accumulation is the main contributing factor to PS1-dependent modulation of CREB signaling.

## APP-dependent enhancement of CREB signaling in neurons expressing FAD-linked PS1 variant

As a read-out of APP-CTF downstream signaling, we first assessed if the expression of PS1 mutation could alter cAMP/PKA-dependent signaling cascades using antibodies that detect phosphorylated PKA substrates (pPKA; *Figure 6—figure supplement 1*). We observed that primary cortical neurons generated from PS1 KI mice exhibit a significant overall increase of phosphorylated PKA substrates (*Figure 6—figure supplement 1*). Associated PKA-dependent CREB signaling was substantiated by an increase in pCREB immunostaining as compared to their WT littermates (*Figure 6a1 and a2*). As we have described previously (*Barnes et al., 2008*), we reiterate here that neurons lacking PS1 function exhibit increased pCREB levels (*Figure 6a2*). As expected for CREB signaling, these effects are more noticeable in the somatic/nuclear area as compared to dendritic projections (*Figure 6—figure supplement 2b*). We confirmed by Western blot analysis that pCREB was significantly increased in neuron lysates generated from PS1 KI mice (*Figure 6b1 and b3*), an effect that was also reported in neuroblastoma cell lines and mouse brains expressing FAD-linked PS1 mutations (*Muller et al., 2011*). In line with PS1 loss-of-function phenotype, Western blot analysis of neuronal lysates confirmed increases of pCREB levels in neurons treated with γ-secretase inhibitor or neurons generated from PS1 KO mouse brains (*Figure 6b2 and b3*). Next, we examined whether or not CREB signaling was influenced by PS1 mutations in cells lacking APP expression. In support of our neurite outgrowth

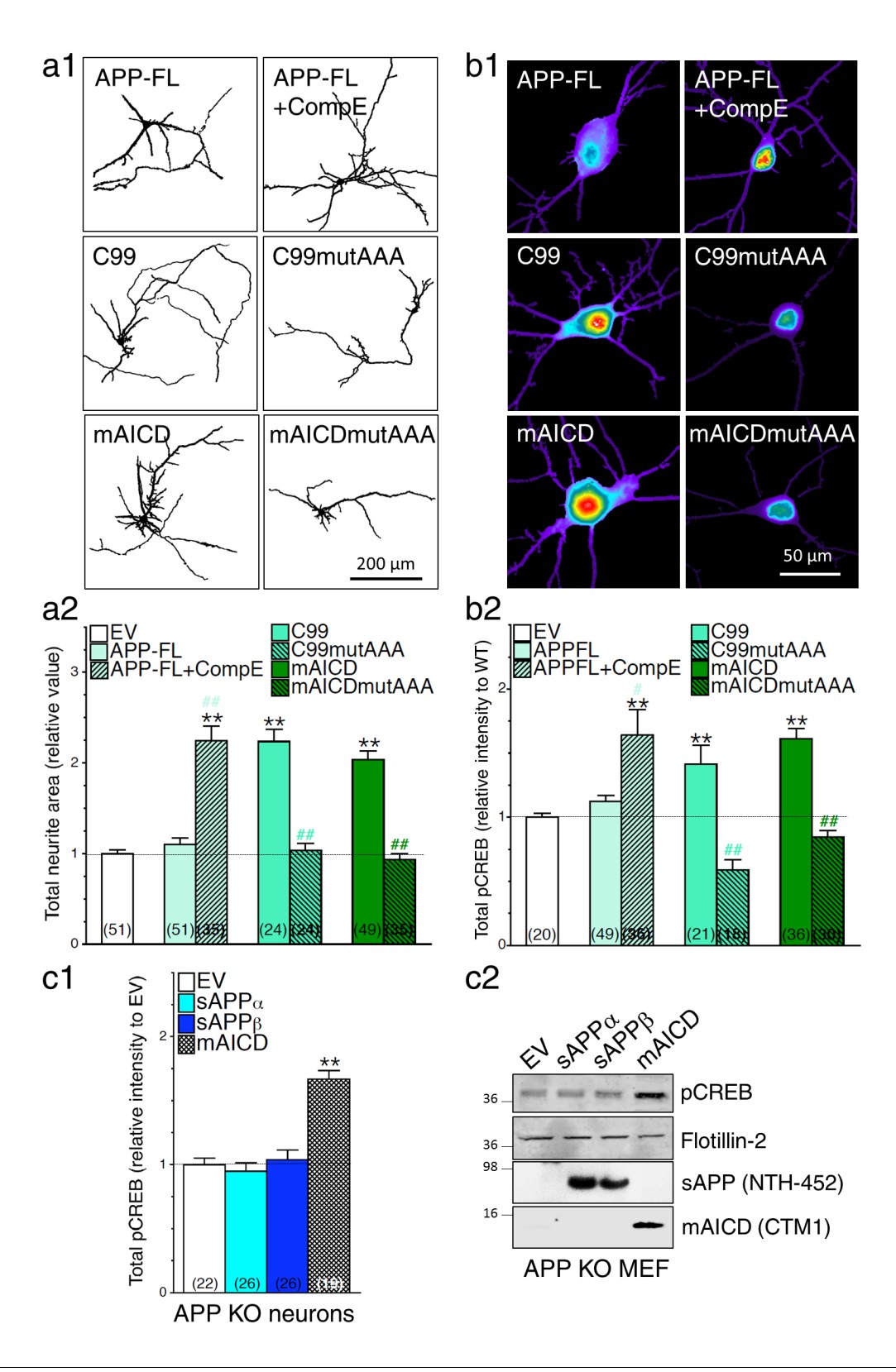

**Figure 5.** Loss of γ-secretase activity is associated with a gain of APP-CTF signaling. (a1) Representative inverted images of cortical neurons (8 DIV) coexpressing YFP and APP-FL, C99, membrane-tethered APP intracellular domain (mAICD) or mutants lacking Gα$_S$-protein interacting site (C99mutAAA

*Figure 5 continued on next page*

*Figure 5 continued*

and mAICDmutAAA) are shown. (**a2**) Analysis of total neurite area is presented as relative to the empty vector (EV) transfected control group. (**b1**) Representative pseudocolor images of phosphorylated CREB (pCREB) immunofluorescence are shown for WT neurons (14 DIV) expressing APP-FL (treated or not with γ-secretase inhibitor Compound E; 10 nM, 24 hr) or various APP-CTF constructs using polyclonal phospho-(Ser$^{133}$) CREB antibody. (**b2**) Quantitative analysis of pCREB staining intensity is represented as relative changes as compared to EV. (**c1**) pCREB immunofluorescence intensity levels were analyzed in 14 DIV neurons expressing EV, soluble APPα (sAPPα), soluble APPβ (sAPPβ) or mAICD that were generated from APP KO mice. Quantitative analysis of pCREB staining intensity is represented as relative changes compared with EV transfected neurons. (**c2**) Western blot analysis of pCREB accumulation is shown in primary mouse embryonic fibroblasts (MEF) expressing EV, sAPPα, sAPPβ or mAICD that were generated from APP KO mice. The level of sAPP was examined in the conditioned media using the NTH-452 antibody that recognized the N-terminal fragment of APP. The expression level of pCREB, Flotillin-2, and mAICD (using CTM1 antibody) were examined in cell lysates. Statistical analysis was performed using ANOVA Kruskal-Wallis test followed by Dunn's post hoc multiple comparison analysis. **$p<0.001$ compared to EV, and #$p<0.05$ and ##$p<0.001$ compared to the overexpressing condition within the same group. The total number of neurons (from at least 3 independent sets of cultures) used for quantification is shown in parentheses. Error bars indicate SEM.

The following figure supplements are available for figure 5:

**Figure supplement 1.** Accumulation of APP-CTF is associated with a larger increase in CREB signaling in the somatic area.

**Figure supplement 2.** Soluble APP does not affect CREB signaling in HEK293 transfected cell lines.

studies presented in *Figure 1*, we likewise observed that neurons generated from APP KO present a significant decrease in pCREB level (*Figure 6a2*). More interestingly, we found that neurons generated from PS1 KI mice that lack APP expression (APP KO x PS1 KI) do not show increased pCREB as observed in PS1 KI littermates, an effect that was not seen in neurons lacking DCC expression (*Figure 6a1, a2, b2, and b3*). Therefore, we conclude that APP expression is associated with the increase in CREB signaling in PS1 KI neurons. Our results confirm that APP is sufficient and necessary to induce CREB signaling in the loss-of-function PS1 mutant (see summary *Table 1*).

## PS1-induced neurite outgrowth and associated CREB signaling require adenylate cyclase activation

CREB signaling possesses a variety of effectors including direct upstream activation of adenylate cyclase and subsequent production of cAMP. We previously reported that APP-CTF induced neurite outgrowth was prohibited in a condition where adenylate cyclase was inhibited (*Deyts et al., 2012*). We documented that APP-CTF initiates stimulation of adenylate cyclase through trimeric interaction with Gα$_S$ (*Deyts et al., 2012*). Therefore, to determine if the increase of CREB signaling seen in PS1 loss-of-function mutants is entirely achieved through the accumulation of APP-CTF mediating adenylate cyclase activation, we treated neurons with adenylate cyclase inhibitor MDL-12,330A (*Figure 7a*). We observed that adenylate cyclase inhibition profoundly reduced CREB activity in neurons generated from PS1 KI. We also confirmed that enhanced neurite outgrowth in neurons expressing PS1 mutants was dependent on adenylate cyclase activity (*Figure 7b*). Moreover, we examined if the FAD-linked mutation in APP could modify the propensity of PS1 mutation to induce neurite outgrowth. We generated cortical neuronal cultures from transgenic mouse embryos coexpressing APPswe and PS1-ΔE9 mutations (Tg85Dbo), a mouse model associated with accumulation of Aβ burden (*Jankowsky et al., 2005*). We observed a significant increase of axodendritic outgrowth in Tg85Dbo as compared to their non-transgenic littermates, at the same level seen in neurons generated from PS1 KO and PS1 KI (*Figure 7b2*). This effect was completely alleviated by a 24 hr treatment with adenylate cyclase inhibitor. Altogether, our results confirm the requirement of cAMP/PKA-dependent signaling to induce neurite outgrowth in neurons that exhibit increased APP-CTF (see summary *Table 1*).

## Discussion

For the past two decades, researchers have focused on the neurotoxicity associated with Aβ peptide production and accumulation. Our study explored the APP molecule as a whole cellular component that could affect neuronal development and perhaps have consequence on the course of AD, thus filling an important gap in AD research. Our results provide a meaningful understanding of how

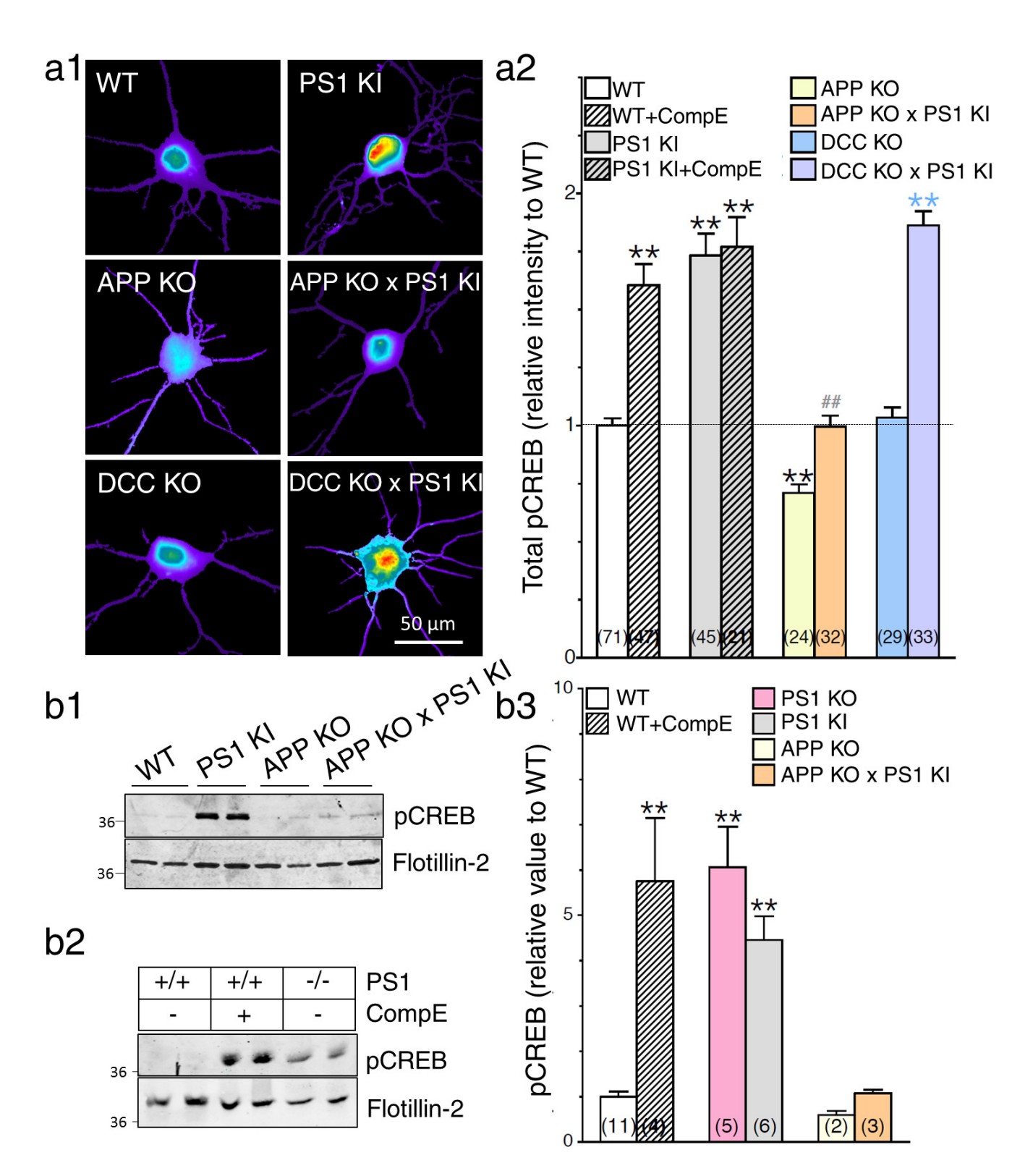

**Figure 6.** APP-dependent enhancement of CREB signaling in neurons expressing FAD-linked PS1 variant. (**a**) Phosphorylated CREB (pCREB) immunofluorescence staining was performed in neurons at 14 DIV using polyclonal phospho-(Ser[133]) CREB antibody. (**a1**) Representative pseudocolor images of pCREB immunostaining levels at steady-state are shown in neurons generated from PS1 KI, APP KO, APP KO x PS1 KI, DCC KO, and DCC

*Figure 6 continued on next page*

*Figure 6 continued*

KO X PS1 KI embryonic brains, and treated with γ-secretase inhibitor Compound E (10 nM, 24 hr). (a2) Quantitative analysis of pCREB staining intensity is represented as relative changes as compared to WT. (b) Steady-state levels of pCREB were examined in neuronal lysates at 14 DIV cortical neurons by Western blot analysis in PS1 KI neurons (b1) and in γ-secretase deficient neurons (b2). Flotillin-2 was used as loading control. (b3) The ratio of pCREB intensity over the intensity of Flotillin-2 as compared to WT is shown. Results were quantified from at least 2 independent cultures. Statistical analysis was performed using ANOVA Kruskal-Wallis test followed by Dunn's post hoc multiple comparison analysis. **p<0.001 compared to WT, blue **p<0.001 compared to DCC KO, and ##p<0.001 compared to PS1 KI. The total number of quantified cells is shown in parentheses (WT, n = 6 embryos; PS1 KI, n = 6 embryos; APP KO, n = 4; APP KO x PS1 KI, n = 4 embryos; DCC KO, n = 4; DCC KO x PS1 KI, n = 7 embryos). Error bars indicate SEM.

The following figure supplements are available for figure 6:

**Figure supplement 1.** Increase in cAMP/PKA signaling in neurons expressing FAD-linked PS1 mutant.

**Figure supplement 2.** The APP-dependent increases of CREB signaling in somatic and dendritic areas.

attenuation of γ-secretase activity and gain of APP function relate to each other, and its significance in health and disease. We took advantage of KI mice expressing FAD-linked PS1 variant, an AD mouse model that does not overexpress APP and PS1 mutations, to demonstrate the necessity of APP and its intracellular metabolites in the development of AD, before the emergence of overt cerebral Aβ burden. We show striking neuronal influences, mechanistic explanations, and in vivo evidence on nerve growth associated with PS1 mutations. Altogether, our findings clearly indicate that APP is an essential substrate in mediating neurite outgrowth in conditions where γ-secretase activity is inhibited or altered. Our results reveal that selective accumulation of membrane-anchor APP-CTF through its coupling to adenylate cyclase is sufficient to induce these changes.

We observed that primary cortical neurons generated from PS1 KO and PS1 KI mice harboring the FAD-linked PS1-M146V variant exhibit exuberant axodendritic outgrowth (see summarized results in *Table 1*). These results correlate with a large and more moderate increase of APP-CTF in brain lysates prepared from PS1 KO and PS1 KI mice, respectively. This supports the notion of partial loss-of-function of γ-secretase activity associated with FAD-linked PS1 mutation. Strikingly, the lack of APP expression in cortical neurons expressing the PS1-M146V variant led to a recovery of basal axodendritic outgrowth, an effect that was not seen in neurons lacking DCC expression (another γ-

**Table 1.** Summary results of characterized mouse models. The mouse breeding to obtain the experimental genotyped, the effect on axodendritic outgrowth, CREB signaling, and APP-CTF accumulation are summarized. Changes are indicated as strongly increase (↑↑), slightly increase (↑), slightly reduce (↓), no-change (-), or not determined (ND). Axodendritic components and associated CREB signaling alterations were evaluated in loss or partial loss of γ-secretase activity using γ-secretase inhibitor Compound E, PS1 KO, APPswe/PS1-ΔE9 (Tg85Dbo) and FAD-linked *PSEN1-M146V* knock-in (PS1 KI) mouse lines crossed to APP KO and DCC KO mice. Axonal as well as dendritic arborizations were increased in cortical neurons generated from PS1 KO, PS1 KI and Tg85Dbo mice. Augmentation of neurite extension correlates with the increase in CREB signaling and APP-CTF accumulation. Lack of APP expression in PS1 KI mice selectively alters axonal outgrowth that parallels a reduction in CREB signaling, as compared to their PS1 KI littermates. Lack of DCC expression in PS1 KI mice does not affect neurite outgrowth and CREB signaling, as compared to their PS1 KI littermates, therefore supporting the idea that APP is an essential substrate, but not DCC, in γ-secretase-mediated axodendritic plasticity.

| Mouse models | Parental strains | Axonal outgrowth | Dendritic outgrowth | CREB signaling | APP-CTF |
|---|---|---|---|---|---|
| Compound E | WT | ↑↑ | ↑ | ↑↑ | ↑↑ |
| PS1 KO | PS1+/- x PS1+/- | ↑↑ | ↑↑ | ↑↑ | ↑↑ |
| PS1 KI/KI | PS1 KI/+ x PS1 KI/+ | ↑↑ | ↑↑ | ↑↑ | ↑ |
| Tg85Dbo | APPswe x PS1-ΔE9 | ↑↑ | ↑↑ | ND | ↑↑ |
| APP KO | APP+/- x APP+/- | ↓ | − | ↓ | none |
| APP KO x PS1 KI | APP+/- x APP+/- PS1 KI/KI | ↓ | − | − | none |
| DCC KO | DCC+/- x DCC+/- | − | − | − | ND |
| DCC KO x PS1 KI | DCC+/- x DCC+/- PS1 KI/KI | ↑↑ | ↑↑ | ↑↑ | ND |

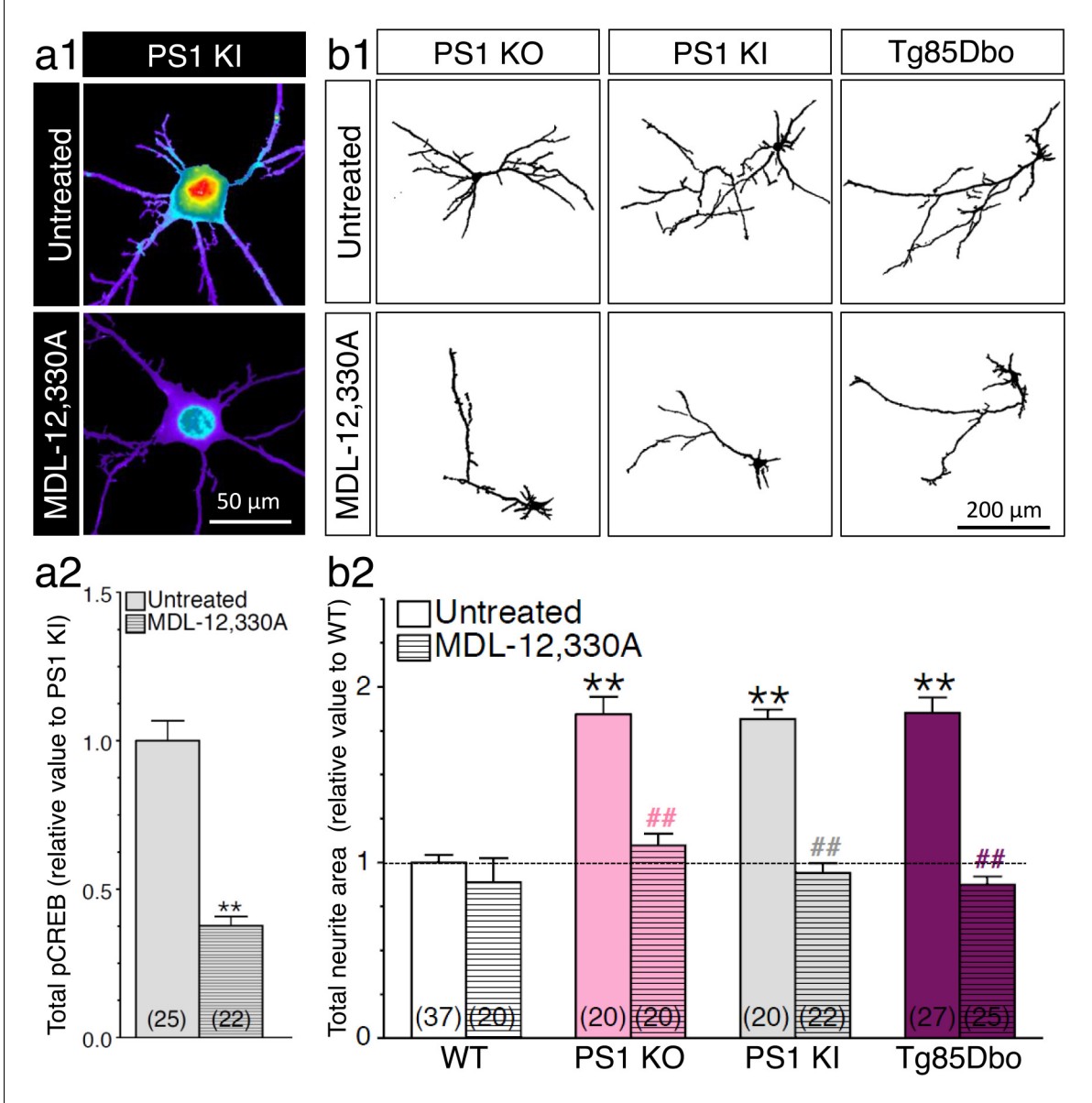

**Figure 7.** PS1-induced neurite outgrowth and associated CREB signaling require adenylate cyclase activation. (**a1**) Representative pseudocolor images of phosphorylated CREB (pCREB) immunofluorescence is shown in 14 DIV neurons generated from PS1 KI mice treated with adenylate cyclase inhibitor MDL-12,330A (100 nM, 30 min) using polyclonal phospho-(Ser[133]) CREB antibody. (**a2**) Quantitative analysis of pCREB staining intensity is represented as relative changes as compared to untreated PS1 KI neurons. **p<0.001 compared to PS1 KI. (**b1**) Representative inverted images of YFP fluorescence in cortical neurons (8 DIV) generated from PS1 KO, PS1 KI, andTg85Dbo mice are shown after 24 hr treatment with MDL-12,330A (10 nM). (**b2**) Analysis of neurite extension is represented as relative changes in total neurite area as compared to WT littermates. **p<0.001 compared to WT, ##p<0.001 compared to untreated condition for each group. The total number of neurons (from at least 3 independent sets of cultures) used for quantification is shown in parentheses (WT, n = 5 embryos; PS1 KO, n = 3 embryos; PS1 KI, n = 3 embryos; Tg85Dbo, n = 3 embryos). Error bars indicate SEM.

secretase substrate). Treatment with the γ-secretase inhibitor did not induce additional axodendritic morphological changes in APP KO x PS1 KI, again supporting the necessity of APP and APP-CTF accumulation in PS1-induced neurite outgrowth. Reduced γ-secretase activity was accompanied by an increase of CREB signaling, an effect that was abolished in APP-deficient neurons and prohibited by adenylate cyclase inhibition. Interestingly, we observed that adult PS1 KI mice exhibit aberrant axonal sprouting, especially in hippocampal areas. More importantly, lack of APP expression in the

FAD-linked PS1 mutant mouse model eliminated that effect, providing evidence that APP expression is causally implicated with the pathological feature associated with loss-of-function PS1 mutation.

Our investigations point out that APP is not an important contributing factor to neurite formation under normal physiological condition. In fact, loss of APP expression only slightly affects axonal outgrowth, at least in cell culture system. Therefore, our observations reinforce the notion that APP must be tightly regulated, likely through secretase-dependent cleavages and degradation process, which would render the molecule less vital for global cellular function. In support of this idea, knocking down APP in mice does not lead to severe phenotype, therefore questioning also the significance of the holoprotein (*Deyts et al., 2016*; *Muller and Zheng, 2012*). Previous studies emphasize this interpretation [review by (*Deyts et al., 2016*; *Haass et al., 2012*; *Muller and Zheng, 2012*; *Thinakaran and Koo, 2008*)]. APP KO mice are viable (*Zheng et al., 1995*), and present subtle, age-dependent decline in memory tasks (*Ring et al., 2007*; *Schrenk-Siemens et al., 2008*). Nonetheless, our study clearly indicates that γ-secretase-dependent inhibition of APP processing, and subsequent APP-CTF accumulation, mediates a striking impact on axodendritic growth. Indeed, we observed that axodendritic outgrowth and associated signaling are reversed in the absence of APP expression in cell culture systems. It has been proposed that ectodomain shedding of cell-surface receptors by γ-secretase complex activity may serve to prevent further cell-surface signaling by a variety of substrates (*Kopan and Ilagan, 2004*; *McCarthy et al., 2009*). Our study clearly demonstrated the importance of this process and the essential contribution of APP-CTF in conditions when PS1 is mutated or γ-secretase activity is reduced. It appears that among all known (and unknown) γ-secretase substrates, APP is the key contributing substrate in observed dysregulation of neurite outgrowth. It is even more remarkable that APP expression would be so critical in controlling neurite development in selective brain areas associated with memory formation. Therefore, APP and its CTF accumulation could play a more critical role in disease states.

Noteworthy, the exuberant axonal sprouting observed in adult PS1 KImice selectively affects the hippocampal areas that are connected with entorhinal cortex/perforant path projections (*Forster et al., 2006*), which is reminiscent of the aberrant sprouting described in AD patients (*Arendt, 2001*; *Geddes et al., 1985*; *Masliah et al., 1991*; *Rekart et al., 2004*). Higher intensity levels of GAP-43 correlated with the hippocampal sprouting in the molecular layer seen in mouse model overexpressing the FAD-linked APP variant (*Chin et al., 2004*; *Phinney et al., 1999*). Mice expressing the FAD-linked PS1 mutation exhibit similitude in that respect. We would consider these sequences of events as an early indication of AD development since Aβ accumulation and deposition have not occurred yet in this model. We corroborated changes seen in our culture systems with that in mouse brains expressing the FAD-linked PS1-M146V variant, reiterating the importance of APP expression as a cause of exuberant neurite outgrowth and sprouting observed in a PS1 KI mouse model. Clearly, it has been reported that lack of APP expression in rodent or *Drosophila* models could lead to inaccuracies in axon guidance in some parts of the brain, including the commissural projections (*Magara et al., 1999*; *Rama et al., 2012*) and mushroom body development, a *Drosophila* brain structure involved in learning and memory (*Soldano et al., 2013*) [see also (*Muller and Zheng, 2012*; *Nicolas and Hassan, 2014*) for review]. More interestingly, *Drosophila* lacking the APP homolog expression requires the APP-CTF component for accurate axonal outgrowth (*Soldano et al., 2013*), indicating the significance of APP-CTF for proper axonal guidance and memory function in *Drosophila*.

One important question that arises from our findings is whether APP/APP-CTF accumulation and associated signaling sequences that impact neurite outgrowth are contributing factors to the disease, or perhaps considered as a compensatory epiphenomenon? One could argue that PS1 loss-of-function mutations favor memory formation since APP-CTF accumulation leads to CREB signaling. It is widely accepted that CREB signaling is important for memory consolidation and cellular events that preserve cell integrity [review by (*Abel and Nguyen, 2008*; *Alberini and Kandel, 2015*; *Lonze and Ginty, 2002*)]. Conversely, ablation of CREB is associated with axonal and dendritic outgrowth defects (*Lonze et al., 2002*; *Redmond et al., 2002*). Therefore, an increase in neurite outgrowth along with CREB signaling would support a prominent role of APP in learning and memory. Even though long-term synaptic plasticity is enhanced in several mouse models expressing FAD-linked PS1 variants (*Parent et al., 1999*; *Parent and Thinakaran, 2010*), which is consistent with an increase in CREB signaling, a number of these mice performed poorly in a variety of memory tasks (*Webster et al., 2014*; *Xia et al., 2015*). One alternative to consider is the detrimental consequence

of over-active CREB signaling. An increase of CREB signaling, in cell lines and brain lysates generated from mouse lines carrying PS1 mutations (including the PS1M146V mutation), can be reversed or normalized by interfering with ER Ca$^{2+}$ stores and the IP$_3$-dependent signaling pathway, an effect that eliminated Aβ toxicity and cell death normally observed in these cells (*Muller et al., 2011*; *Shilling et al., 2014*) [see also (*Guo et al., 1999*)]. Therefore, we stipulate that excessive activation of the CREB pathway may render the cell more vulnerable to injury. Excitatory synaptic plasticity is increased in neurons expressing FAD-linked PS1 variants (*Parent et al., 1999*; *Parent and Thinakaran, 2010*); an effect that causes exaggeration of Aβ production (*Cirrito et al., 2005*; *Kamenetz et al., 2003*). Recent studies demonstrated that chronic entorhinal cortex hyperexcitability in an AD mouse model harboring APP mutations could contribute to a consequential increase in Aβ burden (*Yamamoto et al., 2015*), opening up speculation about the role of APP in the synaptically-driven perforant path cascade of events (*Buxbaum et al., 1998*). Indeed, Aβ burden is reduced in the Tg85Dbo AD mouse model where the perforant path projections are disrupted (*Lazarov et al., 2002*; *Sheng et al., 2002*). It remains to be determined if APP-CTF accumulation would affect any of these outcomes.

γ-secretase dependent accumulation of APP by-products is associated with AD and is considered to be detrimental to neuronal function [review by (*Hardy and Selkoe, 2002*; *Musiek and Holtzman, 2015*)]. Indeed, it has been well documented that the expression of genes encoding several FAD-linked PS1 mutations favors the accumulation of Aβ and potential disease-related toxicity especially in combination with overexpression of APP, at least in animal models (http://www.alzforum.org/research-models). However, a link between AD, aberrant neuronal sprouting in the hippocampal area, partial loss-of-function of γ-secretase activity, and accumulation of APP-CTF has not been described so far. Interestingly, it has been reported that APP-C99 accumulation also takes place selectively in the hippocampal area of 3xTgAD mice (a more aggressive model harboring PS1-M146V/APPswe/Tau-P301L mutations), even prior to accumulation of Aβ (*Lauritzen et al., 2012*). APP-CTF accumulation is found surrounding dystrophic neurites in KI mice expressing the FAD-linked PS1-L166P mutation and two copies of APP-WT (*Vidal et al., 2012*). We also found that neurons generated from transgenic mice coexpressing APPswe and PS1-ΔE9 mutations (Tg89Dbo) exhibit exuberant axodendritic outgrowth that was prohibited by adenylate cyclase inhibition, despite APP-CTF accumulation. Accordingly, we argue that the AD-like aberrant sprouting phenotype depends on signaling downstream of APP-CTF and is preventable through the inhibition of adenylate cyclase activity. Altogether, we demonstrated that APP expression and accumulation of its intracellular fragment are required for exuberant neurite outgrowth associated with pathological PS1 loss-of-function mutations before the emergence of amyloid burden. Our discovery of APP-dependent axonal sprouting in AD mouse models is certainly novel and of great interest for the understanding of how AD process is initiated. Therapeutic inhibition of γ-secretase and the resulting accumulation of APP-CTF could have significant consequences for AD treatment.

## Materials and methods

### Animal models

Mice with targeted deletion of *Psen1* alleles (PS1 KO), targeted deletion of *App* alleles (APP KO), and targeted deletion of *Dcc* alleles (DCC KO) were generated by intercrossing heterozygote PS1 +/- (*Wong et al., 1997*), APP+/- (*Zheng et al., 1995*), and DCC+/- (*Fazeli et al., 1997*), respectively. All these mice were maintained in C57Bl/6J x C3H/HeJ F2 background. Heterozygote transgenic *APPswe/PSEN1-ΔE9* (Tg85Dbo) mouse pairs were purchased from The Jackson Laboratory (Bar Harbor, ME) and maintained in C57Bl/6J background (*Jankowsky et al., 2004*). Homozygote knock-in (KI) mice expressing mutant human *PSEN1-M146V* (PS1 KI/KI) were obtained by intercrossing *PSEN1-M146V* heterozygote (named PS1 KI/+) (*Guo et al., 1999*) and maintained in C57Bl/6J x C3H/HeJ F2 background. Double mutants APP KO x PS1 KI and DCC KO x PS1 KI mice were obtained by crossing heterozygote APP+/- and DCC+/- respectively with heterozygote PS1 KI/+ mice. Animal parental strains are summarized in *Table 1*. Mouse handling procedures were performed in accordance with National Institutes of Health guidelines. The laboratories of Drs. Susan Ackerman (The Jackson Laboratory), Sangram Sisodia (University of Chicago), and Hui Zheng (Baylor

University) kindly provided DCC KO, PS1 KO, APP KO, and PS1 KI original mice for colony expansion.

## Antibodies and reagents

APP-CTM1, APP-CT11, PS1$_{NT}$, Flotilin-2 and APP$_{NTH-452}$ homemade rabbit polyclonal antibodies were generously provided by Dr. Gopal Thinakaran (University of Chicago, Chicago, IL) as described previously (*Deyts et al., 2012*; *Vetrivel et al., 2009*). Monoclonal anti-MAP2, GAP-43 (clone 7B10) and GAPDH were purchased from Sigma-Aldrich (St. Louis, MO). Monoclonal DCC (clone G97-449) antibody was purchased from BD Biosciences (San Diego, CA). Polyclonal phospho-(Ser/Thr) PKA substrate antibody and phospho-CREB (Ser133) antibodies were purchased from Cell Signaling Technology (Danvers, MA) and EMD Millipore (Billerica, MA), respectively. Monoclonal Alexa-647 and Alexa-555, and polyclonal Alexa-555 secondary antibodies were purchased from Invitrogen (Carlsbad, CA). IRDye 680 and IRDye 800CW-conjugated secondary antibodies were purchased from LI-COR Biosciences (Lincoln, NE). γ-secretase inhibitor Compound E was generously provided by Dr. Todd E. Golde (University of Florida, Gainesville, FL) (*Seiffert et al., 2000*). Cis-N-(2-phenylcyclopentyl) azacyclotridec-1-en-2-amine (MDL,12-330A) was obtained from Enzo Life Science (Farmingdale, NY). Tetrodotoxin was purchased from Tocris Bioscience (distributed by Fisher Scientific, Pittsurgh, PA). Unless indicated, all other reagents were purchased from Sigma-Aldrich.

## Cell cultures, expression plasmids, transfections, and treatments

Plasmids encoding empty vector (EV), APP-FL, C99, mAICD, and mAICDmutAAA have been described previously (*Deyts et al., 2012*). Dr. Gopal Thinakaran (University of Chicago, Chicago, IL) generously provided plasmids encoding human PS1-WT, PS1 mutations (PS1-D385A, PS1-M146L, and PS1-ΔE9), and APP mutations (APPswe, APP-F615P, and APP-M596V). cDNAs encoding soluble APP (sAPPα and sAPPβ) and C99mutAAA (RHLSK residues in APP-CTF were mutated to AALSA) were generated by PCR and were cloned into pMX puro retroviral vector. PCR-amplified regions were verified by sequencing. Human embryonic kidney (HEK293), and African monkey kidney (COS-7) cells were originally purchased from the ATCC (Manassas, VA). Stable HEK293 cells overexpressing APP695 were provided from Dr. Sangram Sisodia laboratory. Primary mouse embryonic fibroblasts (MEF), HEK293, and COS-7 cells were cultured in Dulbecco's modified Eagle's medium (DMEM, Invitrogen) supplemented with 10% fetal bovine serum, 1% glutamine and 1% penicillin-streptomycin (Gibco). To avoid mycoplasma contamination, treatment with ciprofloxacin (10 μg/ml) was added once a month for culture maintenance purposes. Cells were used after less than 3–4 passages. Short tandem repeat (STR) profiling was not tested on these well-established cell lines. MEF and HEK293 cells were transiently transfected with Lipofectamine 2000 (Invitrogen) or LipoD293 (SignaGen Laboratories, Rockville, MD), respectively, and according to the manufacturer's protocol. HEK293 cells stably expressing PS1-WT, PS1-D385A, PS1-M146L and PS1-ΔE9 were generated by retroviral infection as described previously (*Deyts et al., 2012*). Briefly, retroviral supernatants collected 48 hr after transfection of Phoenix cells were used to infect HEK293 cells in the presence of 4 μg/ml polybrene. Stably transduced cells were selected in the presence of 1 μg/ml puromycin and pooled for further analysis. Primary mouse cortical neuron cultures were prepared from embryonic E16 mice as previously described (*Parent et al., 2005*) and maintained at 37°C in Minimal Essential Medium (Invitrogen) supplemented with 1% glutamine, 5% horse serum, 0.5% D-glucose, 0.15% HCO$_3$, and nutrients, in a humidified 10% CO$_2$ incubator. Neurons were cultured in 0.1% polyethylenimine-coated 18 mm glass coverslip for immunostaining and poly-L-Lysine-precoated 35 mm dishes for Western blot. Transient transfection of 7 DIV neurons was carried out using Lipofectamine 2000 (Invitrogen) in Neurobasal medium (Invitrogen). After 3 hr, the transfection medium was replaced by 50% original medium and 50% supplemented Minimal Essential Medium without serum. In some cases, cells were treated with Compound E (10 nM) for 24 hr or MDL-12,330A (acute treatment: 100 nM, 30 min; chronic treatment: 10 nM, 24 hr) before being lysed for Western blotting or fixed for fluorescence imaging.

## Immunoblot analysis

Primary neurons or HEK293 cells were lysed in buffer containing 150 mM NaCl, 50 mM Tris-HCl, pH 7.4, 0.5% NP-40, 0.5% sodium deoxycholate, 5 mM EDTA, 0.25% SDS, 0.25 mM

phenylmethylsulfonyl fluoride and protease inhibitor mixture (1:200, Sigma-Aldrich), and briefly sonicated on ice. Mouse brains were lysed in 50 mM NaCl, 25 mM Tris-HCl, pH 7.4, 250 mM sucrose, 1 mM dithiothreitol DTT, 0.25 mM phenylmethylsulfonyl fluoride, protease inhibitor mixture and a phosphatase inhibitor mixture containing the following reagents: 2 mM EGTA, 50 mM NaF, 10 mM Na-pyrophosphate, 20 mM β-glycerophosphate, 1 mM para-nitrophenylphosphate and 0.1 mM ammonium molybdate. Equal amounts of proteins were resolved on SDS-PAGE gels and transferred to a PVDF Immobilon-FL membrane (Millipore, distributed by Fisher Scientific). Endogenous or over-expressed APP full-length (APP-FL) and APP C-terminal fragments (APP-CTF) were detected by immunoblotting with CTM1 antibody. APP-CTFα and APP-CTFβ were separated on 16.5% Tris-Tricine gels. Identification of APP-CTF species was assessed using revelation of APP-CTF from APP-M596V (APP β-site cleavage mutant), APP-F615P (APP α-site cleavage mutant), and APPswe mutant expressing constructs loaded within the same gels as previously described (*Deyts et al., 2012*; *Vetrivel et al., 2011*). Soluble APPs were fractionated on 4–20% SDS gels and revealed using APP$_{NTH-452}$ polyclonal antibody on Western blots made from conditioned media lysates collected 24 hr after transfection (*Vetrivel et al., 2009*). Endogenous DCC full-length (DCC-FL) and DCC endoproteolytic C-terminal fragment (DCC-CTF) were detected using G97-449 DCC antibody. PS1 N-terminal fragment (PS1-NTF) was detected using PS1$_{NT}$ antiserum (*Deyts et al., 2012*). Detection of GAPDH or Flotilin-2 proteins with selective antibodies was used as loading control. Endogenous phosphorylated CREB was detected using pCREB (Ser133) antibody. To increase accuracy and sensitivity, Western blots were quantified by fluorescence using Odyssey infrared imaging system (LI-COR Biosciences, Lincoln, NE). A fixed size box was drawn surrounded the band of interest and quantified within the same gel. Using Odyssey Infrared Imaging software (Li-Cor Biosciences), the band quantification method feature with average background subtraction from top and bottom was employed to determine the level of integrated fluorescence intensity.

## Immunofluorescence

Cells were fixed with 4% paraformaldehyde/4% sucrose in phosphate buffer saline (PBS) for 30 min at 4°C and permeabilized with 0.2% Triton X-100 in PBS for 8 min on ice. After blocking in 10% BSA for 1 hr, neurons were incubated overnight at 4°C with monoclonal MAP2 (1:5000) diluted in 5% BSA in PBS. Then, after washes in PBS, cells were incubated with monoclonal Alexa-647 or polyclonal Alexa-555 conjugated secondary antibodies (1:400) for 1 hr at room temperature and mounted using Permafluor mounting solution (Fisher Scientific). To examine pCREB signaling, cells were serum deprived for 3h in HEPES buffer supplemented with tetrodotoxin (1 μm) as previously described (*Deyts et al., 2012*).

## Image acquisition, processing, and analysis

Labeled neurons were imaged using a motorized Nikon TE 2000 microscope system and Cascade II:512 CCD camera (Photometrics, Tucson, AZ). Using 20X or 60X objective, images were acquired as 200 nm z-stacks and processed using MetaMorph software (Molecular Devices Corporation, Sunnyvale, CA). Total neurite area was evaluated using morphology filter and thresholding features of MetaMorph software to isolate the cell of interest, followed by somatic area removal from YFP expressing cells. The total axonal area was determined by subtraction of total dendritic area (as seen as MAP2 labeling) from total neurite area represented as total pixel number (as seen as YFP fluorescence). To quantify the intensity level of pCREB immunofluorescence, raw images were first set to the same threshold level to eliminate nonspecific fluorescence, and then the gray intensity level was determined and divided by the number of pixels per area (calculated from fluorescence images of YFP transfected cells). In order to evaluate fold of changes between conditions, quantification of area was represented as relative value to the control group. All images were acquired with identical parameters and photomultiplicator values as previously described (*Deyts et al., 2012*).

Labeled neurons were imaged using a motorized Nikon TE 2000 microscope system and Cascade II:512 CCD camera (Photometrics, Tucson, AZ). Using 20X or 60X objective, images were acquired as 200 nm z-stacks and processed using MetaMorph software (Molecular Devices Corporation). Total neurite area was evaluated using thresholding feature of MetaMorph software to isolate the cell of interest, followed by somatic area removal from YFP expressing cells. The total axonal area was determined by subtraction of total dendritic area (as seen as MAP2 labeling) from total neurite area

(as seen as YFP fluorescence). To quantify the intensity level of pCREB immunofluorescence, raw images were first set to the same threshold level to eliminate nonspecific fluorescence, and then the gray intensity level was determined and divided by the number of pixels per area (calculated from fluorescence images of YFP transfected cells). All images were acquired with identical parameters and photomultiplicator values as previously described (*Deyts et al., 2012*).

## Axonal sprouting

Animals were anesthetized with isoflurane and then transcardially perfused with 4% paraformaldehyde/4% sucrose at pH 7.4 dissolved in PBS. Brains were extracted and post-fixed 4-6h in the same solution at 4°C, equilibrated in PBS/30% sucrose for 24 hr, embedded into Tissue-Tek O.C.T. compound (Electron Microscopy Sciences, Hatfield, PA) and then snap-frozen and stored at −80°C until further processing. Coronal sections (40 μm) were performed using a cryostat (Leica CM3050) and stored in a solution containing 30% ethylene glycol, 30% glycerol, and 0.1 M phosphate buffer at −20°C until processing for immunohistochemistry. GAP-43 staining was performed according to the following procedure: brain sections were washed in Tris-buffered saline (TBS, 3 times, 10 min), then blocked for 2 hr in TBS containing 5% horse serum and 0.25% Triton X-100 followed by the incubation with monoclonal GAP-43 (1:500) primary antibody overnight at 4°C. After washes in TBS, sections were then incubated for 2 hr at room temperature with the Alexa-555 secondary antibody (1:400, Invitrogen). Sections were then washed again in TBS before mounting with Vectashield mounting medium (Vector Laboratories, Burlingame, CA). For the quantification, several fields of view per section (every sixth section) were acquired using a motorized Nikon TE 2000 microscope using 4X objective. The immunoreactivity was quantified with Metamorph software by measuring the integrated intensity level divided by the number of pixels per thresholded area. The results were expressed as arbitrary units. Confocal 300 nm z-stack images were acquired using Leica SP5 II STED-CW Super-resolution Laser Scanning Confocal using 20X and 63X objectives. Deconvolved z-stack images were generated using Huygens software (Scientific Volume Imaging, The Netherlands) and processed using MetaMorph software.

## Statistical analysis

Data are presented as mean ± SEM. Statistical significance was determinedby ANOVA Kruskal-Wallis test with independent *post hoc* Dunn's multiple comparisonanalysis using GraphPad prism software (San Diego, CA). Each experiment was performed using, at least, three independent sets of cultures and more than five animals. If statistical significance was not reached among group conditions, a *post-hoc* power analysis was performed to determine needed sample sizes. $*p<0.05$ and $**p<0.001$ as compared to empty vector (EV) transfected cells or WT, and $\#p<0.05$ and $\#\#p<0.001$ as compared to control group within the same experimental conditions.

## Acknowledgements

This work was supported by grants from the National Institutes of Health (NS055223 and AG046710), Bright Focus Foundation, Illinois Department of Public Health, and Alzheimer's Association (IIRG-06-26148) to ATP. We thank Dr. Gopal Thinakaran lab's members, Drs. Mei-ling Joiner and Eric Norstrom for helpful discussions.

## Additional information

### Funding

| Funder | Grant reference number | Author |
| --- | --- | --- |
| BrightFocus Foundation | | Angèle T Parent |
| Illinois Department of Public Health | | Angèle T Parent |
| Alzheimer's Association | IIRG-06-26148 | Angèle T Parent |
| National Institutes of Health | NS055223 | Angèle T Parent |

| National Institutes of Health | AG046710 | Angèle T Parent |

The funders had no role in study design, data collection and interpretation, or the decision to submit the work for publication.

## Author contributions
CD, Conceived and designed the experiments, Performed the experiments, Analyzed the data and wrote the manuscript; MC, SH, NJ, AG, Performed the experiments, Analysis and interpretation of data, Drafting or revising the article; ATP, Conceived and designed the experiments, Analyzed the data and wrote the manuscript, Acquisition of data, Contributed unpublished essential data or reagents

## Author ORCIDs
Angèle T Parent, http://orcid.org/0000-0001-8686-1175

## Ethics
Animal experimentation: This study was performed in strict accordance with the recommendations in the Guide for the Care and Use of Laboratory Animals of the National Institutes of Health. All of the animals were handled according to approved institutional animal care and use committee (IACUC) protocols (#71339) of the University of Chicago. The protocol was approved by the Committee on the Ethics of Animal Experiments of the University of Chicago (Permit Number: A3523-01).

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
