## [Decision Letter]

Thank you for submitting your article "Loss of Presenilin function is associated with a selective gain of APP function" for consideration by *eLife*. Your article has been favorably evaluated by a Senior editor and two reviewers, one of whom, Serge Przedborski, is a member of our Board of Reviewing Editors.

As the Reviewing Editor and one of the Reviewers, I enjoyed reading and reviewing your study a lot and thus would like to invite you to submit a revised version of your work.

For me (Rev #1), the main issue that I would ask you to address throughout is to make it clear how significant and novel your current findings are, i.e. rather than modulating APP which you and others have done before, now you demonstrate striking effects and mechanistic data in the context of PS1 mutation plus the demonstration that some of the nerve growth alterations are detectable in vivo. Honestly, for me, not being an expert in APP and Alzheimer, neither the significance nor the novelty of your new study came across clearly and immediately. As for Reviewer #2, I believe that his/her comments are straightforward.

Reviewer #1 (General assessment and major comments (Required)):

This new manuscript reads quite nicely and reports on a significant topic. The experimental design and methods seem appropriate and the date interpretation and presentation excellent. Despite a high level of enthusiasm for this work, it is not clear to me how novel this work is. It seems that most of the experiments while very well performed and controlled are incremental/confirmatory showing effects of γ secretase on neurite outgrowth via accumulation of APP-CTF and activation of the CERB signaling pathway which the authors or others have found before but this time in the context of the AD-linked PS1 mutations.

Reviewer #2 (General assessment and major comments (Required)):

In this study, Deyts and colleagues set out to investigate the intracellular effects of neurons that carry various forms of mutated Presenilin 1 (PS1), the catalytic subunit of the γ-secretase complex that generates the amyloid β-protein upon its cleavage of the amyloid precursor protein (APP) (as well as investigating the effect of PS1 KO chemically). To achieve this, the authors employed both immortalized cell lines (HEK293) as well as primary cultured neurons from mouse lines expressing various mutated forms of PS1 (M146V and △E9) and the morphology of the neurons are studied microscopically. First, the authors inhibited PS1 pharmacologically and observed an exuberant outgrowth of both axons and dendrites. Interestingly, such effects were not observed in cortical neurons from APP knock-out mice, and they were restored upon addition of an APP-overexpression vectors. From these results, the authors argued that the accumulation of APP intracellular C-terminal fragment (APP-CTF) upon PS1 inhibition is the cause of the outgrowth. Next, the authors observed the same phenotypes when looking at HEK293 cells transfected with vectors carrying different PS1 mutations, all of which required APP expression. To further explore this finding, the authors studied primary cortical neurons from mice carrying the PS1M146V mutation (found in familial cases of Alzheimer's disease – FAD) and found that the increased axodendritic outgrowth is specific to the accumulation of APP-CTF, as knockout of Deleted Colorectal Cancer (DCC), another substrate of PS1, did not abolish the aforementioned phenotype. The authors moved on to study the consequences of neurite outgrowth in vivo by examining the brains of mice expressing the M146V mutant with or without APP expression.

Quantitative analysis of GAP-43 (a marker of axonal plasticity) staining reveals increased intensity in specific regions within the hippocampus, an effect that is not observed in the same mouse strain lacking the APP transgene. Lastly, the authors utilized various biochemical and pharmacological approaches on cultured neurons to demonstrate the CREB signaling cascade and its upstream effector, adenylyl cyclase, as the main mechanism behind the increased axodendritic outgrowth. Overall, the experiments are well-designed and executed and are an important addition to the literature in this area with a few caveats as discussed in minor comments to the authors that should be addressed.

Reviewer #2 (Minor Comments):

1) It is unclear from the Methods section how the axonal area and dendritic area were normalized. Was there a methodological reason for normalizing the area value? It would perhaps be better, if possible to include the actual numbers.

2) In Figure 1C2, it will be helpful to also probe for the C99 fragment, an indication of effective processing by γ-secretase of the βAPP-CTF.

3) In Figure 2A1 and A2, it would be appropriate to include loading controls, particularly for panel A1.

4) Please include images to correspond with the graphs in Figure 2B1 and B2.

5) The Western blot in panel 3B1 would appear to indicate an increase in APP-FL, although the quantification in panel 3B2 would suggest this is not statistically significant. What is the p-value for the comparison ANOVA comparison for the APP-FL expression?

6) In panel 3B1 there appears to be a single band for the APP-CTF in contrast to the two APP-CTF bands corresponding to α and β secretase products observed in panels 3A1and 1C2. Is there an alteration in APP processing in the P10 brains compared to cultured conditions or is the band not able to be resolved?

7) Please include a blot for Figure 3—figure supplement 1.

---

## [Author Response]

For me (Rev #1), the main issue that I would ask you to address throughout is to make it clear how significant and novel your current findings are, i.e. rather than modulating APP which you and others have done before, now you demonstrate striking effects and mechanistic data in the context of PS1 mutation plus the demonstration that some of the nerve growth alterations are detectable in vivo. Honestly, for me, not being an expert in APP and Alzheimer, neither the significance nor the novelty of your new study came across clearly and immediately. As for Reviewer #2, I believe that his/her comments are straightforward.

Reviewer #1 (General assessment and major comments (Required)):

This new manuscript reads quite nicely and reports on a significant topic. The experimental design and methods seem appropriate and the date interpretation and presentation excellent. Despite a high level of enthusiasm for this work, it is not clear to me how novel this work is. It seems that most of the experiments while very well performed and controlled are incremental/confirmatory showing effects of γ secretase on neurite outgrowth via accumulation of APP-CTF and activation of the CREB signaling pathway which the authors or others have found before but this time in the context of the AD-linked PS1 mutations.

We thank the reviewer for his positive remarks. We have addressed this comment and added more justification of the importance of our findings in the Discussion section (first paragraph).

Reviewer #2 (Minor Comments):

1) It is unclear from the Methods section how the axonal area and dendritic area were normalized. Was there a methodological reason for normalizing the area value? It would perhaps be better, if possible to include the actual numbers.

We have decided to illustrate changes based on level of Wt-control condition to facilitate the interpretation of the results between axonal and dendritic growth. This normalization process did not interfere with the significance of the results. For more convincing argument, corresponding raw data of Figure 1 are shown in Figure 1—figure supplement 1. We have also added more describing details in the Methods section (subsection “Image acquisition, processing, and analysis”, first paragraph).

2) In Figure 1C2, it will be helpful to also probe for the C99 fragment, an indication of effective processing by γ-secretase of the βAPP-CTF.

We have added a new supplement figure to illustrate this concept (Figure 1—figure supplement 2). Using HEK293 and COS cells transfected with APP α-site and β-site cleavage mutants, we showed accumulation of β-CTF in γ-secretase inhibited cells. This statement was added in the second paragraph of the Results section.

3) In Figure 2A1 and A2, it would be appropriate to include loading controls, particularly for panel A1.

We have added the corresponding loading control in both panels.

4) Please include images to correspond with the graphs in Figure 2B1 and B2.

We have added the corresponding panels.

5) The Western blot in panel 3B1 would appear to indicate an increase in APP-FL, although the quantification in panel 3B2 would suggest this is not statistically significant. What is the p-value for the comparison ANOVA comparison for the APP-FL expression?

P=0.336 for APP-FL expression. We have added the p-values of ANOVA comparisons as supplement tables (Figure 3—figure supplement 1).

6) In panel 3B1 there appears to be a single band for the APP-CTF in contrast to the two APP-CTF bands corresponding to α and β secretase products observed in panels 3A1and 1C2. Is there an alteration in APP processing in the P10 brains compared to cultured conditions or is the band not able to be resolved?

Western blots illustrated in panels 1C2 and 3A1 were resolved from 16.5% Tris-Tricine gels whereas Western blot from panel 3B1 was resolved from 4-20% SDS gels. We do not have evidence supporting that APP processing is different in P10 brains as compared to cultured conditions (see also Figure 1—figure supplement 2). These precisions were added in the figure legends.

7) Please include a blot for Figure 3—figure supplement 1.

As requested, we have added a blot and showed the quantification taken from immunofluorescence infrared images.